# The Structure Principle and Dynamic Characteristics of Mechanical-Electric-Hydraulic Dynamic Coupling Drive System and Its Application in Electric Vehicle

**Yue Sun** [1,2], **Hongxin Zhang** [1,2,*] **and Jian Yang** [1,2]

1   College of Mechanical and Electrical Engineering, Qingdao University, Qingdao 260071, China; sunyue99213@163.com (Y.S.); yangxiaoming8533@163.com (J.Y.)
2   Power Integration and Energy Storage Systems Engineering Technology Center (Qingdao), Qingdao 260071, China
*   Correspondence: zhx@qdu.edu.cn

**Abstract:** To solve the problem of the low recovery rate of braking energy and the short driving range of electric vehicles, a novel mechanical-electric-hydraulic dynamic coupling drive system (MEH-DCDS) is proposed in this article. MEH-DCDS is a new power integration device that allows electric, mechanical, and hydraulic energy to be converted mutually. It comprises a swash plate plunger pump/motor and a permanent magnet synchronous motor. This article explains the structure and working principles of MEH-DCDS. We describe the dynamic characteristics of MEH-DCDS and analyze the pump and hydraulic motor in the MEH-DCDS hydraulic module. The simulation results show that the flow variation of the MEH-DCDS hydraulic module accords with the design concept of MEH-DCDS, and the pressure variation of high and low pressure accumulators also accords with the theoretical situation. The energy flow of Mechanical-Electric-Hydraulic Power Coupling Electric Vehicle (MEHPC-EV) under different working modes is expounded, and the mathematical model of its key components is established. Based on AMESim and Simulink, the article establishes a vehicle simulation dynamic model. The dynamic performance of MEHPC-EV in UDDS is analyzed by co-simulation. The simulation results show that the application of MEH-DCDS in electric vehicles is feasible. MEHPC-EV reduced battery energy consumption by 26.18% compared to EV. The research in this paper verifies the accuracy and superiority of the system, which has a significant reference value for the development and study of electric vehicles in the future.

**Keywords:** mechanical–electric–hydraulic; co-simulation; dynamic characteristics; AMESim; simulink

## 1. Background and Significance

### 1.1. Introduction

The current energy structure of the world is dominated by fossil energy such as coal, oil, and natural gas, which are non-renewable and will produce toxic substances after burning. In this era, the development of pure electric vehicles is conducive to reducing the world's dependence on non-renewable energy and can play a role in protecting the environment [1,2]. Automotive companies are also increasingly focusing on the development of electric and hybrid electric vehicles [3,4]. In addition, pure electric vehicles and hybrid electric vehicles have good dynamic performance and can quickly recover braking energy [5]. Sharifan [6] used ADVISOR software to study the application performance of asynchronous motors and permanent magnet motors in hybrid electric vehicles. The simulation results show that the permanent magnet motor has better efficiency-torque-speed characteristics. The electrical losses of the PM motor are much lower than the induction motor during the drive-cycle of the hybrid electric vehicle. This research reveals some advantages of the permanent magnet motor compared with the asynchronous motor in the hybrid electric vehicle. Wang [7] proposed an outer-rotor permanent-magnet synchronous

motor with a novel U-shape rotor structure. The simulation results show that the new U-type permanent magnet motor improves torque and reduces cogging torque. Jung-woo Park [8] introduced a method to improve the control characteristics of built-in permanent magnet synchronous motors in electric vehicles. At the same time, the torque reference distribution law is proposed to improve the driving characteristics of the two kinds of electric vehicles. The numerical calculation and experiment verify the superiority of the method. Because the permanent magnet synchronous motor has the advantages of high power density and high reliability [9,10], this article chooses it as the core component of the mechanical-electric-hydraulic dynamic coupling drive system (MEH-DCDS) electric energy conversion system.

Ma [11] designed a new electro-hydraulic braking system based on regenerative braking energy and established a corresponding mathematical model. The simulation results show that the liquid capacity of the regenerative brake accumulator is a crucial factor affecting energy recovery efficiency. His research can provide an analytical reference for the practical application of the electro-hydraulic system. Li [12] proposed a new type of electro-hydraulic braking system. The system integrates the characteristics of regenerative braking control and anti-lock control. The simulation results show that the integrated electro-hydraulic braking system is suitable for energy recovery and emergency braking. Yang [13] designed a new type of electro-hydraulic hybrid drivetrain. The results show that the new electro-hydraulic hybrid drive system can effectively avoid the influence of a sizeable charge-discharge current on battery cycle life. While showing good dynamic performance, vehicle economy has improved significantly. Li [14] presented the mathematical model of a hydraulic accumulator and preliminarily determined its basic working parameters. The rationality of the operation parameters of the accumulator was verified by the test bed. The study helps people to understand the characteristics of accumulators. Niu [15] introduced an electric hydraulic hybrid vehicle. In ideal circumstances, the battery energy consumption of urban delivery trucks is reduced by 30% on average, and the mileage of urban buses is extended by 50%. The system reduces the power consumption of the electric powertrain. Wang [16] designed a new energy-saving hydraulic system, which uses a regeneration unit composed of a hydraulic motor and a generator to replace the traditional pressure compensator. The simulation results show that the system has good control performance and a remarkable energy-saving effect.

Chen [17] compared the energy efficiency of hydraulic hybrid vehicles and hybrid electric vehicles. The simulation results show that the fuel consumption of the hydraulic hybrid vehicle is lower than that of the hybrid electric vehicle. In addition, hydraulic hybrid vehicles consume less electricity than pure electric vehicles. The research shows the good performance of the hydraulic hybrid vehicles. Rezaei [18] designed a new adaptive energy consumption minimization strategy (ECMS) algorithm for energy management of plug-in hybrid electric vehicles. Based on this algorithm, Rezaei proposed an energy consumption minimization strategy-catch energy saving opportunity (ECMS-CESO) adaptive law for plug-in hybrid electric vehicles. Using the proposed adaptive law, the optimal value of the ECMS equivalence factor can be estimated at any time for both charge-depletion and charge-sustaining modes.

Battery packs in pure electric cars are big and heavy. The power of pure electric vehicles and series hybrid electric vehicles is entirely provided by the motor installed on the vehicle [19,20]. Therefore, this needs to be powered by high-power motors. Although parallel hybrid electric vehicles have two driving systems, engine and motor, the motor is still needed as the main power source in some working modes [21]. Therefore, this also requires a large installed power of the motor. When these cars brake, the motor regains some of the braking energy. However, limited by the generator's maximum output and the battery's maximum charging efficiency, the energy recovered by electric regenerative braking is small [22].

In this article, a new mechanical–electric–hydraulic dynamic coupling drive system (MEH-DCDS) is designed to achieve the mutual transmission of electric energy, mechanical

energy, and hydraulic energy. MEH-DCDS combines a permanent magnet synchronous motor with a swash plate plunger pump/motor. The device overcomes the shortcomings of a traditional electro-hydraulic system, such as its loose structure and single working mode. Its application in an electric car can be a Mechanical-Electric-Hydraulic Power Coupling Electric Vehicle (MEHPC-EV). It can use hydraulic power to efficiently recover braking energy, which can significantly improve the driving range of electric vehicles and reduce the peak torque of electric vehicles. At the same time, the vehicle's dynamic property and fuel economy have been effectively improved.

### 1.2. Challenges of the MEH-DCDS

The research on MEH-DCDS and MEHPC-EV is still in the simulation stage and has not been verified by relevant experiments. The MEH-DCDS prototype needs to be processed and mounted on electric vehicles. It is also necessary to conduct real vehicle experimental research on MEHPC-EV.

### 1.3. Contribution of This Paper

(1)  In this paper, a new mechanical-electro-hydraulic dynamic coupling drive system (MEH-DCDS) is proposed, which consists of a permanent magnet synchronous motor and a swash plate plunger pump/motor. It can realize the mutual conversion between mechanical energy, hydraulic energy, and electrical energy.
(2)  The dynamic characteristics of MEH-DCDS are analyzed.
(3)  MEH-DCDS was applied to electric vehicles, and the feasibility of MEHPC-EV was verified by co-simulation.

### 1.4. The Structure of the Paper

(1)  Section 2 illustrates the structure of MEH-DCDS.
(2)  Section 3 introduces the working principle of MEH-DCDS.
(3)  Section 4 completes the mathematical model of MEH-DCDS.
(4)  Section 5 analyzes the dynamic characteristics of MEH-DCDS.
(5)  Section 6 describes the application of MEH-DCDS in electric vehicles.
(6)  Section 7 realizes the simulation and analysis of MEHPC-EV.

## 2. The Structure of MEH-DCDS

Four systems of MEH-DCDS are designed to realize the conversion of mechanical energy, hydraulic energy, and electric energy, including supporting system, electric energy conversion system, hydraulic energy conversion system, and mechanical energy conversion system.

### 2.1. Supporting System

The main structure of the support system is the shell, and the shell is the frame of the MEH-DCDS. The internal capacity of the shell is large, which can accommodate the components of the other three systems. The shell is provided with a hole in the center of the upper end. An annular groove is provided on the inner wall of the lower end of the hole, used to place the sealing ring. The drive shaft extends from the hole in the housing. The lower part of the shell is a bolted back cover, which is provided with a high pressure oil mouth and a low pressure oil mouth. The two oil ports are connected with the waist-shaped hole on the valve plate. The housing is equipped with bearing bushes that support the hydraulic energy conversion system. There is also a fixed stator core structure inside the shell.

### 2.2. Electric Energy Conversion System

The core of the electric energy conversion system is the stator and rotor parts of the motor. The motor of this system adopts a three-phase permanent magnet synchronous motor, which has the characteristics of small volume, high power density, high reliability,

and high efficiency [23,24]. The rotor center meshes with the drive shaft, and the drive shaft rotates the rotor. There are six holes centered around the center hole, which are used to place the plunger. When moving, the rotor is subjected to great centrifugal force, which requires the rotor core to have high mechanical strength. The stator core is made of cold-rolled silicon steel sheets, which are insulated from each other and contain high silicon. The use of a sheet iron core makes eddy currents pass through a small cross-section in a long and narrow circuit, thus increasing the resistance of the eddy path. At the same time, the silicon in the silicon steel increases the resistivity of the material and also plays a role in reducing the eddy current. The stator iron core is evenly arranged around three groups of stator windings with a chain structure. Each coil of the chain windings has the same size, which is easy to manufacture.

### 2.3. Mechanical Energy Conversion System

The mechanical energy conversion system mainly comprises the drive shaft, bearing, sealing ring, flat washer, stop spline sleeve, and other components, which can realize the mechanical energy input and output of MEH-DCDS. The stop spline sleeve plays a role in limiting the piston pump cylinder block to move along the drive shaft direction. The front bearing plays a role in limiting the drive shaft to move up along the axis. The drive shaft is stepped, and the shoulder of the shaft plays the role of axial positioning here. The middle part of the drive shaft can be matched with the internal spline shaft sleeve of the axial piston pump so that the drive shaft can rotate synchronously with the cylinder block of the hydraulic system. The drive shaft extends out from the shell of the support system, and the load motor is connected externally to realize the internal and external output of mechanical energy.

### 2.4. Hydraulic Energy Conversion System

The hydraulic energy conversion system mainly comprises plunger, swash plate, slide shoe, valve plate, spring, etc. These hydraulic components can form a plunger-cylinder pair, slipper-swash plate pair, cylinder-valve plate pair, and other motion pairs. These motion pairs cooperate to achieve the conversion between hydraulic energy, mechanical energy, and electrical energy.

The working stroke of the plunger is determined by the working angle of the swash plate. The inclined direction of the swash plate determines the movement direction of the plunger along the axial direction, and the movement direction of the plunger determines the flow direction of the hydraulic oil at the high and low pressure oil mouth. The spring and rear bearing confine the drive shaft to moving down the axis. Valve plate plays the role of high and low pressure oil distribution. The hydraulic oil of the plunger is supplied and unloaded through the waist hole on the valve plate. The guide sleeve is embedded in the first half of each plunger hole and has a guiding sealing effect on the plunger. The bottom of each plunger hole and the center of the whole plunger are provided with an oil hole. When the plunger rotates obliquely and reciprocates, the oil can flow into or out of the plunger center. The four major system structures of MEH-DCDS are shown in Figure 1.

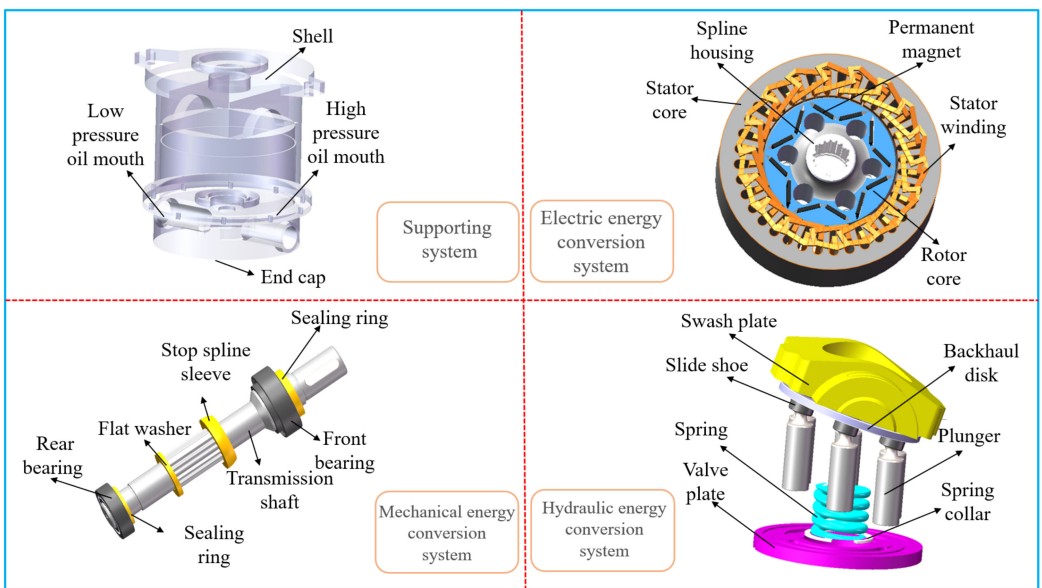

**Figure 1.** Four major system structures of the mechanical-electric-hydraulic dynamic coupling drive system (MEH-DCDS).

## 3. The Working Principle of MEH-DCDS

MEH-DCDS integrates a swash plate plunger pump/motor and permanent magnet synchronous motor. It combines the high precision of motor drive with the high power density of hydraulic drive. The basic working principle of MEH-DCDS is the mutual conversion of mechanical energy, electrical energy, and hydraulic energy [25,26]. According to the different use modes, it can realize single-to-single energy conversion and meet the requirements of multi-to-single or single-to-multi energy conversion. Therefore, it has a broad application prospect.

When mechanical energy and electrical energy are converted into each other, MEH-DCDS can be understood as a permanent magnet synchronous motor. When operating as a generator, the prime mover turns the drive shaft. The rotor rotates accordingly because the drive shaft meshes with splines on the rotor. The permanent magnet, whose magnetic field rotates with the rotor, senses the electromotive force by cutting the three-phase winding of the stator. The conversion of mechanical energy to electrical energy can be realized by leading the alternating current with a wire. When the MEH-DCDS works as an electric motor, the lead wire is externally connected to three-phase AC. The rotating magnetic field generated by the armature winding of the stator acts on the rotor to generate torque. The torque drives the rotor to rotate so that the drive shaft outputs mechanical energy outward with the rotation of the rotor. It has realized the transformation from electrical energy to mechanical energy.

When mechanical and hydraulic energy is converted into each other, MEH-DCDS can be understood as swash plate plunger pumps/motors. When MEH-DCDS operates in the form of a plunger pump, the prime mover drives the drive shaft to rotate. The drive shaft drives the rotor to rotate by the spline meshing with the rotor. The plunger in the plunger hole of the rotor does rotation with the rotor on the one hand. On the other hand, under the action of mechanical devices such as swash plate and slipper and oil pressure do reciprocate linear movement in the cylinder block. When the plunger extends, the volume of the inner cavity formed by the plunger and the plunger hole increases. At the moment, the hydraulic oil flows from the low pressure accumulator into the low pressure oil mouth of the MEH-DCDS. It then enters the plunger cavity through the waist hole on the valve plate to complete the oil absorption process. When the plunger is squeezed into the cavity, the plunger and plunger hole formed by the cavity volume decreases. The hydraulic oil goes through the plunger cavity from the waist hole on the plate into the high pressure

accumulator. MEH-DCDS completes the oil discharge process. When the rotor rotates a circle, each plunger has experienced a reciprocating movement, so each plunger cavity has also experienced the process of oil absorption and oil discharge, causing the mechanical energy to hydraulic energy conversion. When MEH-DCDS works in the form of a hydraulic motor, the hydraulic oil flows from the high pressure accumulator into the high pressure oil mouth of MEH-DCDS, and enters the plunger cavity through the waist hole on the valve plate. The plunger extends under the action of high pressure oil. Torque is formed by oil pressure in interaction with the plunger and swash plate. The torque can drive the rotor and drive shaft rotation when the plunger rotates near the low pressure oil mouth. At this point, the plunger squeezes into the cavity, and hydraulic oil goes through the plunger cavity from the waist hole on the plate into the low pressure accumulator. When the rotor rotates a circle under the action of hydraulic oil, each plunger experiences a reciprocating movement. Each plunger experiences the process of oil absorption and oil discharge, which can realize hydraulic energy conversion to mechanical energy.

When electrical and hydraulic energy are converted to each other: When the lead of MEH-DCDS is externally connected to three-phase AC, the windings on the stator generate a rotating magnetic field, which is applied to the rotor to create torque. The torque will drive the rotor rotation, the plunger on the one hand with the rotor to make the rotating motion, on the other hand in the cylinder block to make the reciprocating linear motion. The movement of the plunger will generate hydraulic energy, which realizes the conversion of electrical energy to hydraulic energy. When the high pressure oil flows from the high pressure accumulator into the plunger cavity, the plunger extends under the action of the high pressure oil. The oil pressure interacts with the plunger and swash plate to create a torque that drives the rotor. The permanent magnetic field cuts the stator windings as the rotor rotates. The alternating electromotive force is induced in the three-phase windings, thus realizing hydraulic energy conversion to electric energy. The energy conversion diagram of MEH-DCDS is shown in Figure 2.

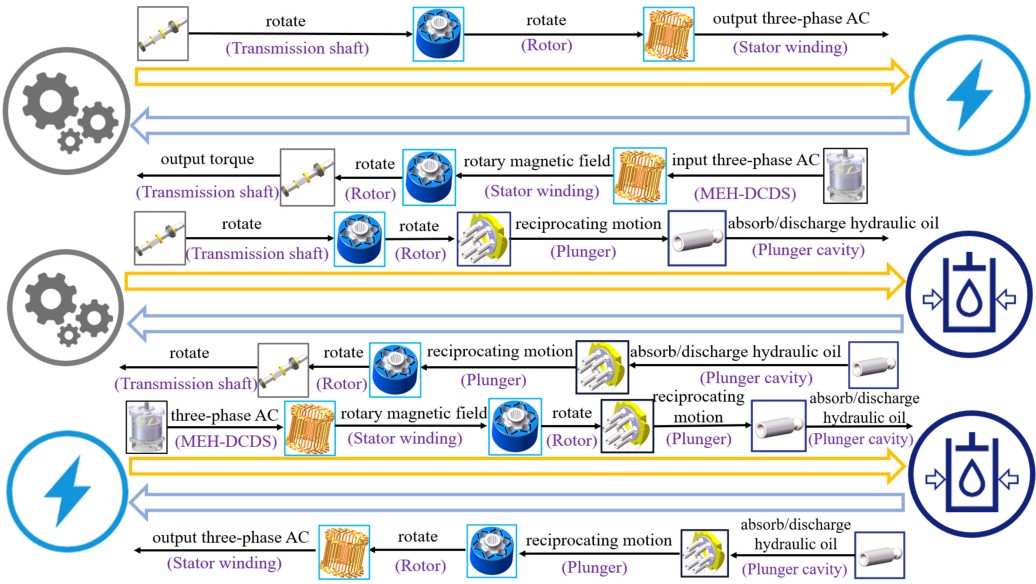

**Figure 2.** Energy conversion diagram of MEH-DCDS.

## 4. Mathematical Model of MEH-DCDS

MEH-DCDS integrates a swash plate plunger pump/motor and permanent magnet synchronous motor. Therefore, an electricity power analysis and hydraulic power analysis of the MEH-DCDS was performed. Hydraulic power analysis is based on the hydraulic pump/motor, and electric power analysis is based on the motor and battery. This section primarily establishes the hydraulic and electric power mathematical models.

### 4.1. Hydraulic Power

When the car is turned on, the hydraulic system works alone. The hydraulic motor spins because of the hydraulic oil. The hydraulic motor transforms hydraulic energy into mechanical energy to drive the car. When the vehicles accelerate, the battery transmits electrical energy to the MEH-DCDS, and the hydraulic and electric power jointly drive the car [27]. When the vehicle is braking, the hydraulic pump stores the excess braking energy in the form of hydraulic energy in the hydraulic accumulator.

The hydraulic pump/motor's output torque $T_p$ is calculated as follows:

$$T_p = \frac{\Delta_p V_p \beta}{2\pi} \eta_p \tag{1}$$

In the formula, $\beta$ is the swash plate opening, $(-1,1)$; $\Delta_p$ is the pressure difference between the high pressure accumulator and low pressure accumulator; $V_p$ is the displacement of hydraulic pump/motor; and $\eta_p$ is the mechanical efficiency of hydraulic pump/motor.

The flow $Q$ of hydraulic pump/motor is calculated as follows:

$$Q = \frac{\pi}{4} d^2 Rz \tan\gamma \tag{2}$$

where $d$ is the diameter of the plunger; $R$ is the diameter of the plunger cavity; $z$ is the number of plungers; and $\gamma$ is the inclination angle of the swash plate.

The hydraulic accumulator has the function of storing braking energy, which is mainly used to start the car. Hydraulic power has a higher power density than batteries and can start the car quickly.

The relationship between volume and pressure can be determined by Boyle's law, which can be calculated as follows:

$$p_0 v_0{}^n = p_1 v_1{}^n = p_2 v_2{}^n \tag{3}$$

where $p_0$ is the pre-charge pressure of the gas; $v_0$ is the accumulator charging volume; $p_1$ is the minimum working pressure of the accumulator; $v_1$ is the volume of gas before the accumulator works; $p_2$ is the maximum working pressure of the accumulator; and $v_2$ is the gas volume after the accumulator works.

The hydraulic accumulator has three modes. The dynamic computation is based on these three modes when the hydraulic accumulator works.

(a) If the accumulator's energy is sufficient, the accumulator's gas pressure is maximum, and the gas volume is minimal, therefore:

$$P_{gas} = P_{\max} \tag{4}$$

$$V_{gas} = \frac{V_0}{1000} \tag{5}$$

where $P_{gas}$ is the gas pressure; $P_{\max}$ is the maximum pressure; $V_{gas}$ is the volume of gas; and $V_0$ is the volume of the accumulator.

(b) If the energy of the accumulator has been fully released, the gas pressure of the accumulator is the pre-charge value, and the volume of the accumulator is the maximum, therefore:

$$P_{gas} = P_0 \tag{6}$$

$$V_{gas} = V_0 \tag{7}$$

(c) If the hydraulic accumulator is neither fully discharged nor fully charged, the air pressure is the same as the hydraulic pressure, hence:

$$P_{gas} = P_{out} \tag{8}$$

where $P_{out}$ is the output pressure of the accumulator.

The volume of the gas is calculated as follows:

$$V_{gas} = V_0 \cdot \left( \frac{P_0}{P_{gas}} \right)^{\frac{1}{y}} \tag{9}$$

*4.2. Electric Power*

In MEH-DCDS, the motor recovers braking energy, and the input torque $T_{req}$ of the motor is confined by the limit torque $T_{lim}$:

$$T_{min} \leq T_{lim} \leq T_{max} \tag{10}$$

where $T_{lim}$ is the minimum torque of the motor; and $T_{max}$ is the maximum torque of the motor.

The motor's output torque $T_m$ is dictated by the limit torque $T_{lim}$, which is calculated using first-order lag.

$$T_m = \frac{1}{1 + t_r \cdot s} \cdot T_{lim} \tag{11}$$

where $t_r$ is the user-defined time constant.

The mechanical power $P_{mec}$ and power loss $P_{lost}$ of the motor are calculated as follows:

$$P_{mec} = \frac{T_m \cdot n_e}{9549} \tag{12}$$

$$P_{lost} = (1 - \eta_e) \cdot |P_{mec}| \tag{13}$$

where $n_e$ is the rotational speed of the motor shaft.

The relationship between mechanical power $P_{mec}$ and electrical power $P_{elec}$ is as follows:

$$P_{elec} = P_{mec} - P_{lost} \tag{14}$$

The efficiency of motor/generator:

$$\eta_m = 2 - \frac{P_{elec}}{P_{mec}} \tag{15}$$

$$\eta_g = \frac{P_{elec}}{P_{mec}} \tag{16}$$

where $P_{elec}$ is the electric power; $\eta_m$ is the motor efficiency; and $\eta_g$ is the generator efficiency.

The battery is the power source that drives the system. The output voltage $U_{out}$ and $SOC$ of the battery are calculated as follows:

$$U_{out} = U_{oc} - IR \tag{17}$$

$$SOC(t) = SOC_0 - \frac{\int_0^t I(t)dt}{Q_0} \tag{18}$$

where $U_{oc}$ is the open-circuit voltage of the power battery; $R$ is the internal resistance of the power battery; $I$ is the current of the power battery; $SOC$ is the state of charge of the power battery; $SOC_0$ is the initial state of charge of the power battery; and $Q_0$ is the capacity of the power battery.

## 5. The Dynamic Characteristics of MEH-DCDS

MEH-DCDS is used in electric cars. Before the MEH-DCDS can be installed on electric vehicles, a hydraulic dynamic analysis of the MEH-DCDS is required. We regard MEH-DCDS as a hydraulic pump/motor for dynamic analysis of its hydraulic part. If the

hydraulic characteristics of MEH-DCDS are reasonable, we will install MEH-DCDS on electric vehicles to analyze whether the vehicle simulation model is correct.

### 5.1. Analysis of Pump in MEH-DCDS Hydraulic Module

When the hydraulic module of MEH-DCDS operates as a hydraulic pump, the external torque drives the drive shaft. The oil flows out of the low pressure accumulator through the low pressure oil area of the valve plate into the cylinder block. When the cylinder block rotates to the high pressure oil area of the valve plate, the oil flows from the cylinder block to the high pressure accumulator under the force of the swash plate and the plunger. The plunger goes in and out once a cycle.

The analysis model of the MEH-DCDS hydraulic module pump is shown in Figure 3. The driving module outputs speed, and drives the hydraulic module of MEH-DCDS to rotate by the moment of inertia. The high and low pressure accumulators are connected to the high pressure and low pressure oil ports of MEH-DCDS, respectively. A certain running state of the MEH-DCDS hydraulic module is simulated to verify the feasibility of the model. Setting the speed as 1400 r/min, the initial pressure of the high pressure accumulator is 27 MPa, and the initial pressure of the low pressure accumulator is 18 MPa. The speed of the hydraulic module is 1400 r/min, so the working cycle is 0.043 s. That is to say, for every 0.043 s, each plunger chamber has completed a process of oil absorption and oil discharge.

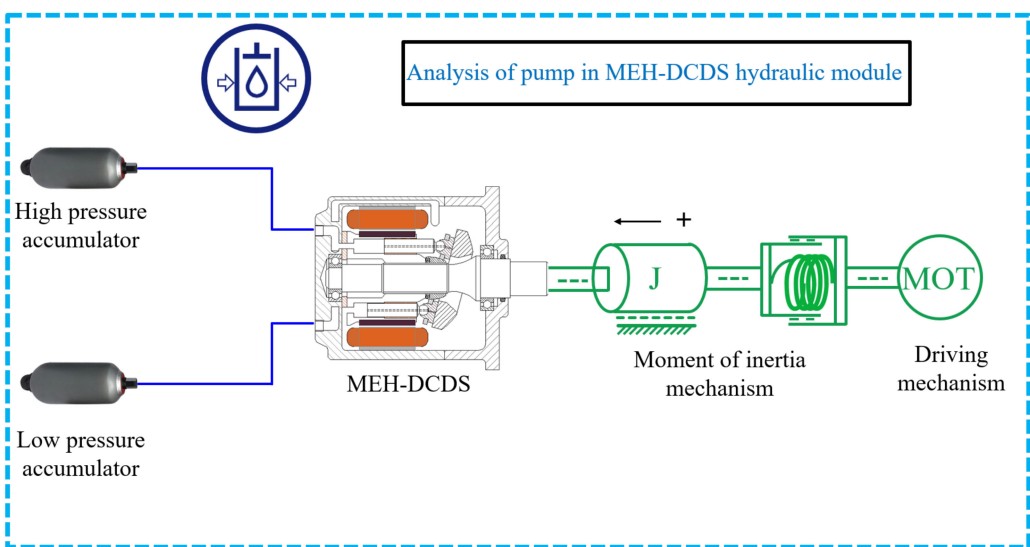

**Figure 3.** Analysis model diagram of pump in MEH-DCDS hydraulic module.

According to the positive direction of the model, the flow out of the hydraulic module is positive, and the flow into the hydraulic module is negative. There are six plungers in MEH-DCDS, and we selected the plunger just in the oil suction stage for analysis. At the beginning of the plunger chamber for the oil suction stage, hydraulic oil from the low pressure accumulator goes into the plunger chamber, and the pressure of the low pressure accumulator is reduced. There must be another plunger in the discharge stage. The hydraulic oil enters the high pressure accumulator from the plunger chamber, making the pressure of the high pressure accumulator rise. The plunger chamber we used for analysis was in the low pressure oil region at the beginning of the simulation, when hydraulic oil flowed from the low pressure accumulator into the plunger chamber. So, from 0 s to 0.0214 s, the plunger has a flow change at the low pressure port. For the high pressure oil port, the plunger is also in the low pressure oil area, so the plunger flow in the high pressure oil port is 0. When the simulation reaches 0.0214 s, the plunger just rotates to the high pressure oil port for the oil discharge process. At this time the plunger in the high pressure oil port flow changes. Similarly, the flow rate of the plunger at the low pressure

oil port is 0. Therefore, from 0.0214 s to 0.043 s, the plunger has a flow change at the high pressure oil port, while the flow is 0 at the low pressure oil port. At this point, the plunger just completed one cycle of work. In the figure, flow pulsation occurs when the oil inlet and oil outlet are converted into each other, which is caused by flow backflow when high and low pressure oil are converted into each other. Figure 4 shows the pressure variation of the low pressure accumulator (LPA) and the flow variation of the plunger at the low pressure oil port. Figure 5 shows the pressure variation of the high pressure accumulator (HPA) and the flow variation of the plunger at the high pressure oil port.

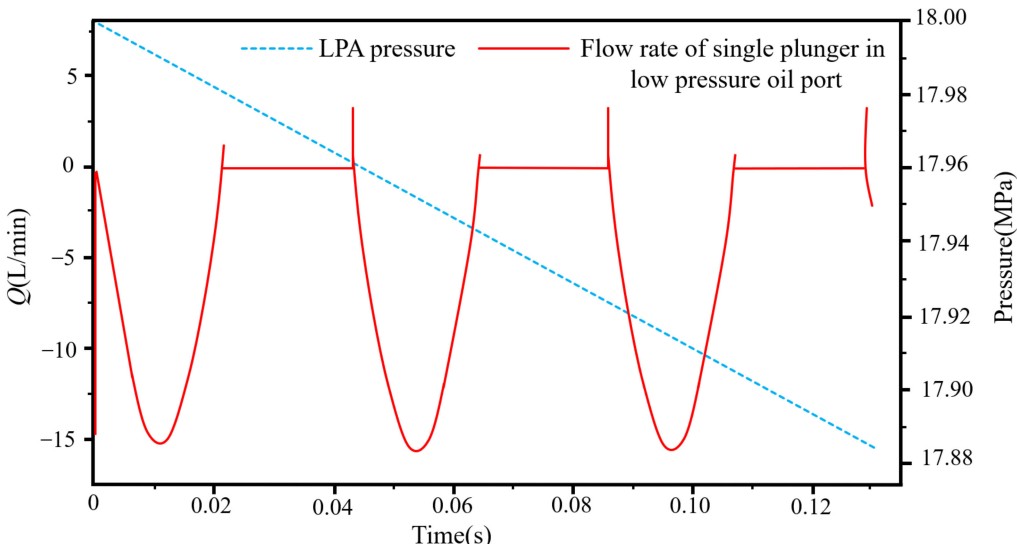

**Figure 4.** Diagram of low pressure accumulator (LPA) pressure and single plunger flow rate at low pressure oil port on pump analysis.

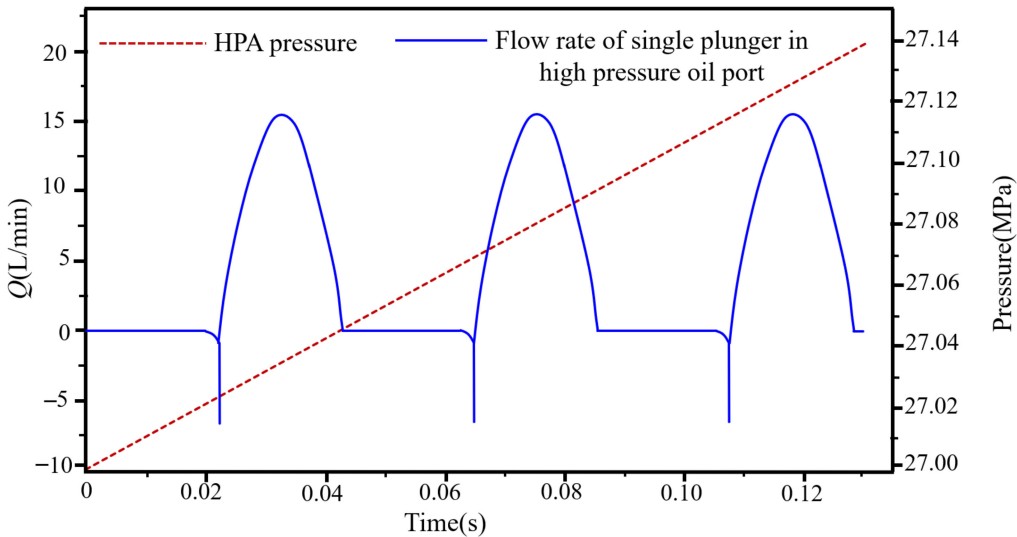

**Figure 5.** Diagram of high pressure accumulator (HPA) pressure and single plunger flow rate at high pressure oil port on pump analysis.

### 5.2. Analysis of Motor in the MEH-DCDS Hydraulic Module

For the simulation of the motor in the MEH-DCDS hydraulic module, we can change the driving mechanism in the hydraulic module model into a loading mechanism, so that the pump model can be changed into a motor model. The simulation model is shown in Figure 6. A certain working state of the motor is simulated to verify the feasibility of the model. We set the high pressure accumulator to 35 MPa, the low pressure accumulator to

10 MPa, and then applied an external load. The simulation analysis of the motor in the hydraulic module can then be carried out.

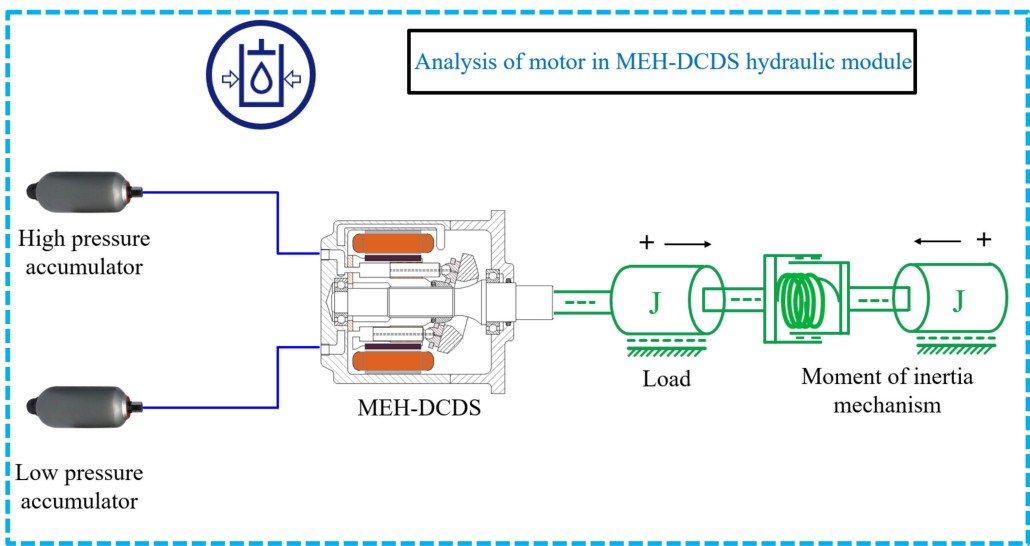

**Figure 6.** Analysis model diagram of the motor in MEH-DCDS hydraulic module.

The analysis of the motors is similar to that of the pump, but the individual plunger we choose to analyze is different. The hydraulic motor can output torque, which is an energy release mechanism. In the pump analysis, we chose the plunger that had just rotated to the low pressure oil port, while in the motor analysis we chose the individual plunger that was just in the discharge stage, that is, the plunger that had just been rotated to the high pressure oil port.

When the plunger is just in the oil discharge stage, the hydraulic oil enters the plunger chamber from the high pressure accumulator, so that the pressure of the high pressure accumulator is reduced. At this point, there will be another plunger just in the oil suction stage, pushing hydraulic oil from the plunger chamber into the low pressure accumulator, causing the pressure of the low pressure accumulator. The plunger chamber we analyzed was in the high pressure oil region at the beginning of the simulation, and the hydraulic oil flowed into the plunger cavity from the high-pressure accumulator. So, in the beginning, the piston pump has a flow change at the high pressure oil port, and the flow is 0 at the low pressure oil port. When the plunger pump rotates to the low pressure oil port, the plunger pump has a change in the low pressure oil port flow, and the high pressure oil port flow is 0. This is similar to pump analysis. There is a large flow pulsation when the inlet and outlet flow are converted to each other, which is caused by the shock of pressure mutation when the high and low pressures are transformed. Figure 7 shows the pressure variation of the high pressure accumulator (HPA) and the flow variation of the plunger at the high pressure oil port. Figure 8 shows the pressure variation of the low pressure accumulator (LPA) and the flow variation of the plunger at the low pressure oil port.

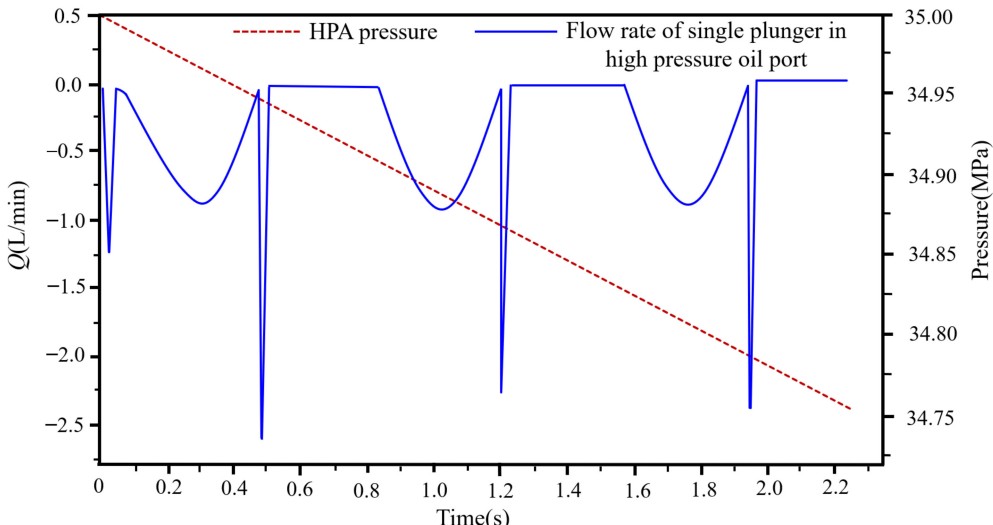

**Figure 7.** Diagram of HPA pressure and single plunger flow rate at high pressure oil port on motor analysis.

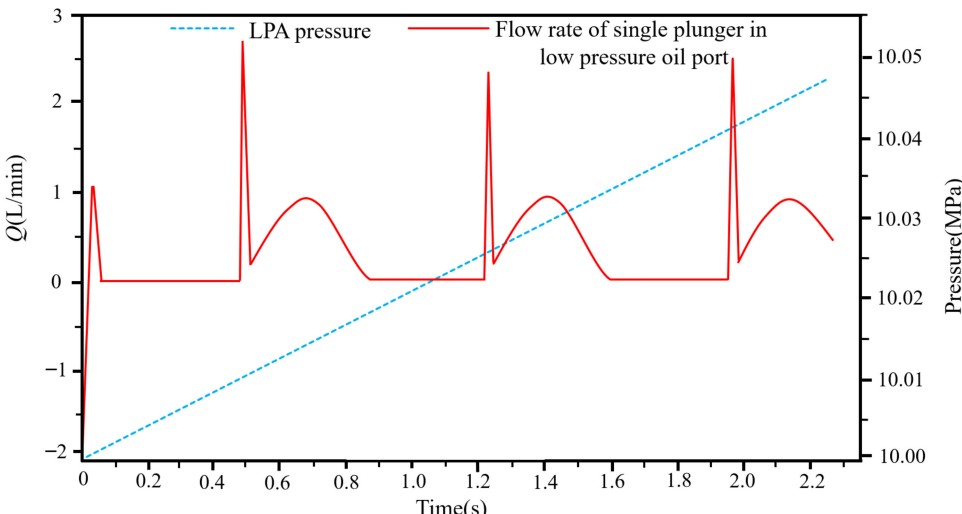

**Figure 8.** Diagram of LPA pressure and single plunger flow rate at low pressure oil port on motor analysis.

## 6. Application of MEH-DCDS in Electric Vehicles

When conventional electric vehicles are running, a change in load produces an electric shock. This kind of shock can cause damage to the battery and electric control system. The braking energy recovery process of the pure electric vehicle is limited by the maximum power generation and maximum charging power of the battery pack, so energy recovery efficiency is low. Some commercial electric vehicles use hydraulic booster pumps as auxiliary power, but their structure is loose, and the control is complicated [28].

MEH-DCDS can be applied to electric vehicles to form MEHPC-EV, which overcomes the limitations of traditional electric vehicles. When MEHPC-EV brakes, the hydraulic system and motor work together to recover braking energy. The hydraulic module recovers the braking energy and stores it in the high pressure accumulator, and the energy recovered by the motor module is stored in the power battery pack. Through reasonable control of the system, the energy recovery advantages of the electric energy module and the hydraulic energy module can be fully combined, which significantly improves the energy recovery efficiency of the automobile [29].

MEHPC-EV is the main application direction of MEH-DCDS. The efficient working area of the hydraulic pump/motor and motor is not the same. Therefore, this article adopts different working modes according to different operating conditions of automobiles. This approach can give full play to the advantages of MEH-DCDS. According to the structural characteristics of MEH-DCDS, MEHPC-EV can be divided into four basic operating modes: starting mode, acceleration mode, constant speed mode, and braking mode.

When the traditional electric vehicle starts, the motor produces a large current. Such a large current has a great influence on the battery state of charge(SOC)and increases the loss of electric energy. Thus, in starting mode, MEHPC-EV only has a hydraulic system to drive the car. In acceleration mode, the power battery provides electricity to MEH-DCDS, and the high pressure accumulator provides hydraulic energy to MEH-DCDS. Both electrical and hydraulic energy provides power to the car at the same time, which greatly improves the power performance of the car. In constant speed mode, the hydraulic system does not work and only the motor drives the car. Motors usually operate within the range of high efficiency. In braking mode, the hydraulic and electromagnetic modules of the MEH-DCDS can recover braking energy simultaneously. The hydraulic module converts part of the braking energy into hydraulic energy and stores it in the high pressure accumulator. The electromagnetic module converts the other part of the braking energy into electricity and stores it in the battery pack. This energy recovery method obviously improves the energy recovery efficiency. Figure 9 shows the structure of MEHPC-EV and the energy flow of MEHPC-EV under different working modes.

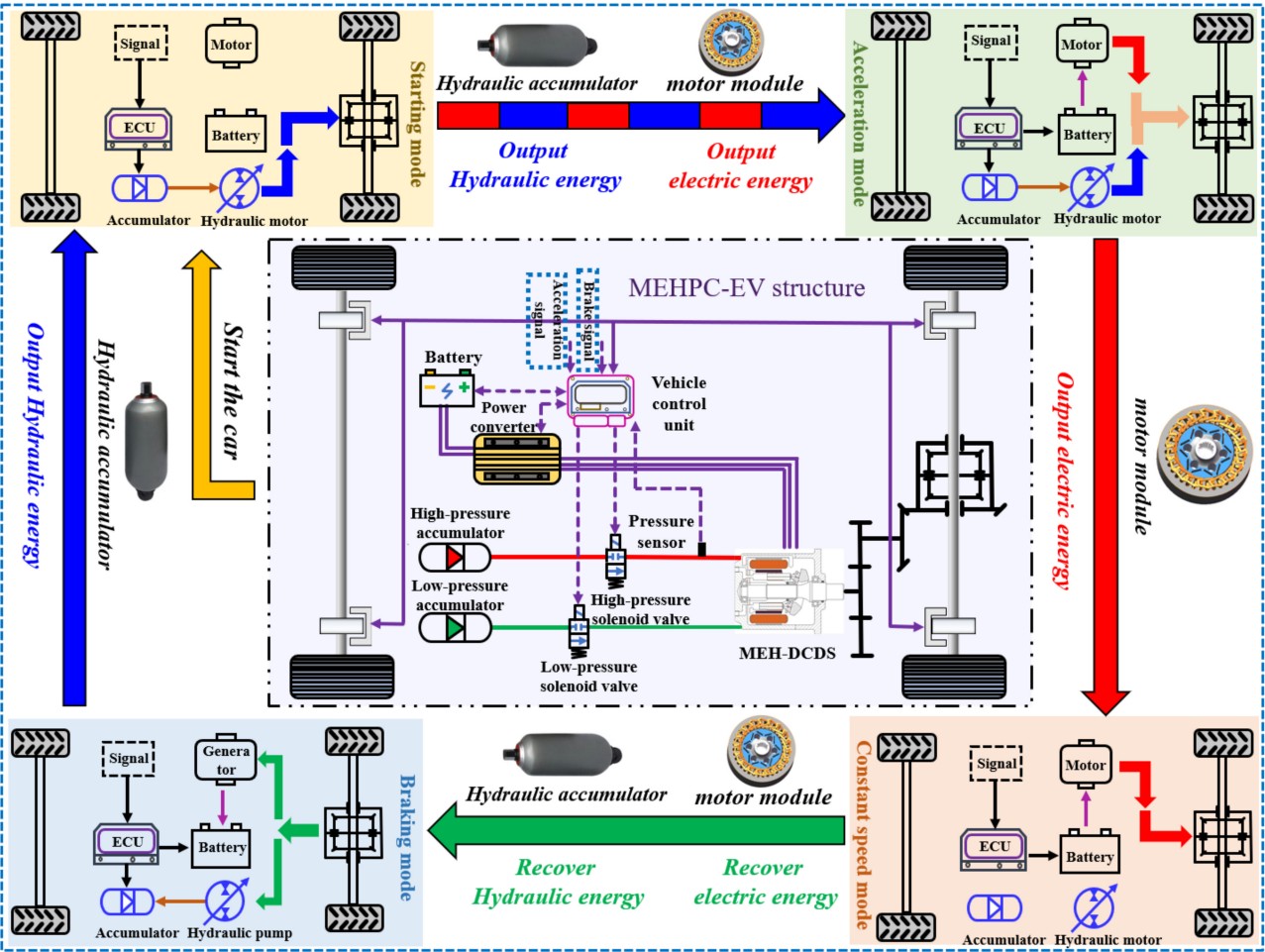

**Figure 9.** Diagram of Mechanical-Electric-Hydraulic Power Coupling Electric Vehicle (MEHPC-EV) structure and energy flow.

## 7. Simulation and Analysis of MEHPC-EV

### 7.1. Whole Vehicle Simulation Model

MEHPC-EV parameters are determined according to the vehicle's dynamic model requirements. Its basic parameters are shown in Table 1.

**Table 1.** Main parameters of the MEHPC-EV.

| Components | Main Parameters | Value |
|---|---|---|
| | Loaded mass (m) | 1206 kg |
| | Frontal area (A) | 2.28 m$^2$ |
| Main parameters of the car | Rolling resistance coefficient ($f$) | 0.0135 |
| | Coefficient of air resistance ($C_D$) | 0.32 |
| | Wheel width (R) | 290 mm |
| | Transmission efficiency ($\eta$) | 0.85 |
| High pressure accumulator | Work pressure ($P_1$) | 240–350 Bar |
| | Volume (V) | 35 L |
| Low pressure accumulator | Work pressure ($P_2$) | 60–220 Bar |
| | Volume (V) | 35 L |
| Secondary component | Displacement ($V_P$) | 30 mL·r$^{-1}$ |
| Motor | Rated power ($P_e$) | 32 KW |

In this paper, the vehicle simulation model is built based on AMESim and Simulink. AMESim is the authoritative software for electro-hydraulic hybrid electric vehicle simulations. Simulink is used to build rule-based control strategies. Through the method of co-simulation, the whole vehicle model is built. The essential modules in this model include the high pressure accumulator module, low pressure accumulator module, battery module, and MEH-DCDS module. In AMESim, the MEH-DCDS module integrates secondary components (hydraulic pump/motor) and a permanent magnet synchronous motor.

A module of the secondary component can only work in one mode at a time. The module will work as a hydraulic motor if the hydraulic pump/motor is selected to give a negative command. Figure 10 shows the whole vehicle simulation model.

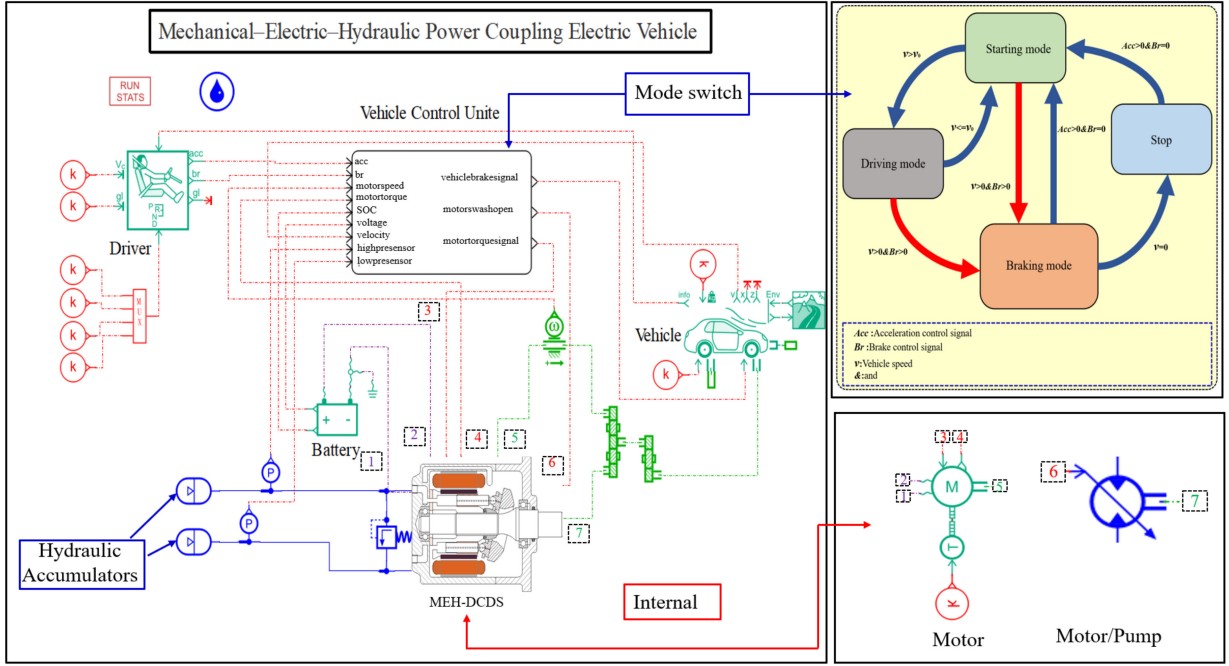

**Figure 10.** Whole vehicle simulation model.

*7.2. Results and Analysis*

UDDS (Urban Dynamometer Driving Schedule) is the urban road cycle condition for passenger cars and light trucks. As a result, this article uses UDDS to validate the driving conditions of the MEHPC-EV. The feasibility of MEHPC-EV is verified by co-simulation of AMESim and Simulink.

Figure 11 shows the simulation analysis of MEHPC-EV. The MEHPC-EV's real speed overlaps with the control speed without serious deviation. This indicates that the path tracking accuracy is high. This strategy can meet the expectation of actual driving conditions and has certain reliability. As the HPA pressure increases, the LPA pressure decreases. As LPA pressure goes up, HPA pressure goes down. This is consistent with the working philosophy of MEH-DCDS and proves the feasibility of MEHPC-EV. Compared to electric vehicles (EV), MEHPC-EV has a significantly lower peak motor torque and significantly higher energy recovery. This verifies the feasibility and superiority of MEHPC-EV in terms of motor peak torque characteristics and energy recovery. The figure shows the MEHPC-EV and EV battery SOC consumption rate curve. The simulation results show that the remaining battery SOC of EV is 92.9828%, and that of MEHPC-EV is 94.8201%. The energy consumption rate is reduced by 26.18%.

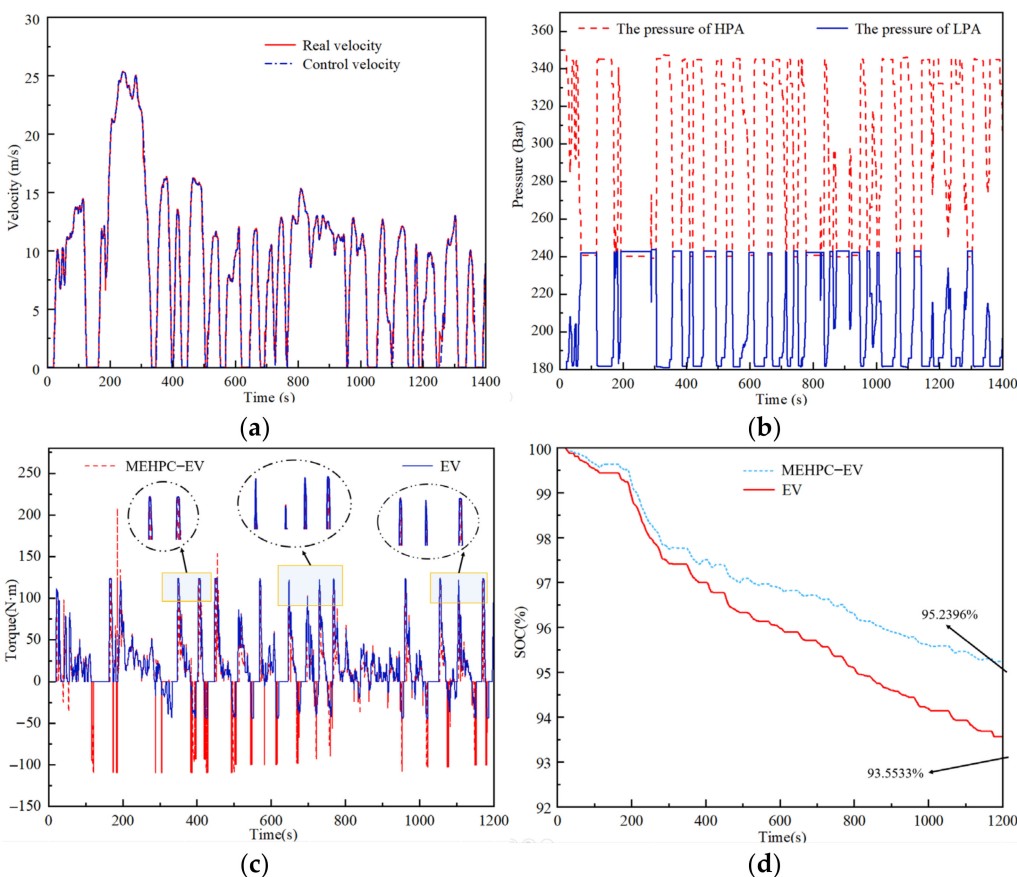

**Figure 11.** Diagram of MEHPC-EV simulation analysis. (**a**) is the speed curve, (**b**) is the accumulator pressure curve, (**c**) is the motor torque curve, and (**d**) is the battery SOC consumption rate curve.

## 8. Conclusions

This article proposes a new mechanical-electric-hydraulic dynamic coupling drive system (MEH-DCDS). It combines a permanent magnet synchronous motor with a swash plate pump/motor to achieve the mutual transformation of electric energy, mechanical energy, and hydraulic energy. Firstly, this paper designs four MEH-DCDS systems and introduces their structure and function. According to the design characteristics of MEH-

DCDS, the process and principle of energy conversion are described in detail. Then, the mathematical models of hydraulic and electric power of MEH-DCDS are established. The dynamic characteristics of MEH-DCDS are also analyzed. In the pump analysis and motor analysis of the hydraulic module, the plunger that has just been rotated to the low pressure oil port and the high pressure oil port are selected for analysis. respectively. The change of flow rate of the single plunger in the high and low pressure orifice accords with the design concept. The pressure variation of high and low pressure accumulators also accords with the theoretical situation of MEH-DCDS. In order to give full play to the advantages of MEH-DCDS, this paper classifies the working modes of MEHPC-EV and studies the energy flow under different working modes of MEHPC-EV.

In this article, the simulation model of MEHPC-EV is built based on AMESim and Simulink. Finally, the dynamic performance of MEHPC-EV under the UDDS condition is analyzed by using co-simulation method. The results show that MEHPC-EV's real speed can follow the control speed well. The pressure variation of high and low pressure accumulators also conforms to the design concept. Compared with EV, the peak motor torque of MEHPC-EV is significantly reduced, and energy recovery is significantly improved. MEHPC-EV achieved a 26.18% reduction in battery energy consumption compared to EV.

The new electro-hydraulic hybrid structure proposed in this study solves the problems of low energy recovery efficiency and short driving range of electric vehicles. This provides a significant idea for further studying the electro-hydraulic hybrid power system.

**Author Contributions:** Conceptualization, Y.S. and H.Z.; methodology, J.Y.; formal analysis, J.Y. and H.Z.; investigation, Y.S. and H.Z.; resources, Y.S. and H.Z.; writing—original draft preparation, Y.S.; writing—review and editing, Y.S. and H.Z. All authors have read and agreed to the published version of the manuscript.

**Funding:** This research was funded by the National Natural Science Foundation of China, grant number 52075278, and Municipal Livelihood Science and Technology Project of Qingdao, grant number 19-6-1-92-nsh.

**Conflicts of Interest:** The authors declare no conflict of interest.

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
