# Peer review of "The Structure Principle and Dynamic Characteristics of Mechanical-Electric-Hydraulic Dynamic Coupling Drive System and Its Application in Electric Vehicle"

_electronics, doi:10.3390/electronics11101601_

Round 1

Reviewer 1 Report

The author has proposed a novel mechanical-electric-hydraulic dynamic coupling drive system for improving low recovery from the braking system of Electric vehicles. The research paper is interesting. However, the reviewer has the following concerns:

  1. Clearly identify and write the major contributions of the research paper in the introduction section.
  2. Include more test cases to verify the efficacy of the proposed method.
  3. Even though the energy consumption rate is reduced by 26.18%. However, no comparisons have been provided with already available schemes. 
  4. Carefully proofread the paper.

Author Response

Response to Reviewer 1 Comments

Point 1: Clearly identify and write the major contributions of the research paper in the introduction section.

Response 1: I added this paper's contribution to the paper. Section 1.3 is the contribution of this paper.

Point 2: Include more test cases to verify the efficacy of the proposed method.

Response 2: I added the analysis of the dynamic characteristics of MEH-DCDS. The section 5 of the paper is the analysis of the dynamic characteristics of MEH-DCDS. This part adds to the persuasive power of the approach.

Point 3: Even though the energy consumption rate is reduced by 26.18%. However, no comparisons have been provided with already available schemes.

Response 3: I have made an explanation in the Word file submitted.

Point 4: Carefully proofread the paper.

Response 4:I have gone over the revised paper carefully. Some sentences have been revised to make them more streamlined.

Reviewer 2 Report

The paper topic is interesting and relevant. The results of the presented studying describe the proposed strategy, which can meet the expectation of actual driving conditions and has certain reliability. The main benefit of research is real speed overlaps with the control speed without serious deviation.
But, suggesting some improvements to this paper:
1. Fig.1 and Fig.2 are similar, so maybe use one.
2. Fig.7 is not visible, because is very small, dont see the main blocks.
It may be possible to simplify some part or highlight some part in other figures, For example, subsystems of the Simulink model is not possible to read.
3. Fig.9 has poor quality, please improving.

Author Response

Response to Reviewer 2 Comments

Point 1: Fig.1 and Fig.2 are similar, so maybe use one

Response 1: I deleted Figure 1. I think what the reviewer said is reasonable. Figure 1 and Figure 2 are repeated.I have explained it in the word file I have submitted.

Point 2: Fig.7 is not visible, because is very small, dont see the main blocks.

It may be possible to simplify some part or highlight some part in other figures, For example, subsystems of the Simulink model is not possible to read.

Response 2: I modified the picture to make the switching between different working modes more clear.I have explained it in the word file I have submitted.

Point 3: Fig.9 has poor quality, please improving.

Response 3: I have made major changes to this article. I think this article should strengthen the description of MEH-DCDS and cut some mehPC-EV content. Therefore, I added the analysis of the dynamic characteristics of MEH-DCDS in the section 5 and deleted some parts about the control strategy. Therefore, I deleted Figure 9 and added the mode switching part in the vehicle simulation model diagram.I have explained it in the word file I have submitted.

Reviewer 3 Report

Good quality paper, but there are some formal improvement that can be done like label in picture 4 are over line and are hard to read and label in picture 7 are small.

Author Response

Response to Reviewer 3 Comments

Point 1: Good quality paper, but there are some formal improvement that can be done like label in picture 4 are over line and are hard to read and label in picture 7 are small.

Response 1: I have modified figures 4 and 7 to make sure figure 7 looks clean.

Reviewer 4 Report

After studying the article, there are the following notes:

1.Provide a list of contribution(s) of the manuscript. The properties mentioned in line 88-90 are desired properties of any algorithm. Also, consider providing a brief outline of the manuscript at the end of Section 

2.Literature review is not complete. For instance, there are many control schemes that could be used to improve the literature review. For instance, consider discussing the following manuscripts :

a)K. Gao, Z. Meng, K. Jiang, H. Zhang and Q. Zeng, "Shearer Height Adjustment Based on Mechanical-Electrical-Hydraulic Cosimulation," in IEEE Access, vol. 8, pp. 222064-222076, 2020, doi: 10.1109/ACCESS.2020.3043516.

b)S. Rezaee, E. Farjah and B. Khorramdel, "Probabilistic Analysis of Plug-In Electric Vehicles Impact on Electrical Grid Through Homes and Parking Lots," in IEEE Transactions on Sustainable Energy, vol. 4, no. 4, pp. 1024-1033, Oct. 2013, doi: 10.1109/TSTE.2013.2264498.

c)A. Emadi, Y. J. Lee and K. Rajashekara, "Power Electronics and Motor Drives in Electric, Hybrid Electric, and Plug-In Hybrid Electric Vehicles," in IEEE Transactions on Industrial Electronics, vol. 55, no. 6, pp. 2237-2245, June 2008, doi: 10.1109/TIE.2008.922768.

d)A. Rezaei, J. B. Burl, M. Rezaei and B. Zhou, "Catch Energy Saving Opportunity in Charge-Depletion Mode, a Real-Time Controller for Plug-In Hybrid Electric Vehicles," in IEEE Transactions on Vehicular Technology, vol. 67, no. 11, pp. 11234-11237, Nov. 2018, doi: 10.1109/TVT.2018.2866569.

3.What would be the main problems of real implementation of proposed system (or method)? Please explain and elaborate.

4.What is the difference between this paper and these works ([19], [25] and [30])?

5.References must be written according to the journal's policy. Follow this example in writing references:

  1. Eltamaly, A.M., Al-Saud, M., Sayed, K., Abo-Khalil, A. G. Sensorless active and reactive control for DFIG wind turbines using opposition-based learning technique. Sustainability, 2020, 12, 3583. https://doi.org/10.3390/su12093583.
  2. Benbouhenni, H.; Bizon, N. Terminal Synergetic Control for Direct Active and Reactive Powers in Asynchronous Generator-Based Dual-Rotor Wind Power Systems. Electronics 2021, 10, 1880. doi: 10.3390/electronics10161880.

Author Response

Response to Reviewer 4 Comments

Point 1: Provide a list of contribution(s) of the manuscript. The properties mentioned in line 88-90 are desired properties of any algorithm. Also, consider providing a brief outline of the manuscript at the end of Section .

Response 1: I added some content. Section 1.3 is the contribution of this paper. Section 1.4 is the structural outline of this paper.

Point 2: Literature review is not complete. For instance, there are many control schemes that could be used to improve the literature review. For instance, consider discussing the following manuscript.

Response 2: I carefully read the papers provided by reviewers, and selected helpful papers to quote. References cited are [3],[4],[18].

Point 3: What would be the main problems of real implementation of proposed system (or method)? Please explain and elaborate.

Response 3: Since MEH-DCDS is still in the simulation stage and has not been processed and manufactured, we can only analyze the possible problems in the actual implementation of MEH-DCDS from the aspects of structure and principle. Because MEH-DCDS can realize the mutual conversion of three kinds of energy and the structure is highly integrated, the failure of any part of the mechanical part, hydraulic part, and motor part will have a significant impact on the whole system. For example, if a permanent magnet is subject to thermal demagnetization, not only the electrical power will be affected, but also the hydraulic and mechanical power will be greatly affected. In other words, MEH-DCDS has high integration and advantages.,but if a part of MEH-DCDS goes wrong, it will have a huge impact on the rest of the structure.

Point 4: What is the difference between this paper and these works ([19], [25] and [30])?

Response 4:Literature [19] is a review paper.The characteristics and typical models of energy sources of pure electric vehicles are firstly described.  Then the existing pure electric  vehicle types are depicted and the environmental impacts of the typical pure electric vehicles are evaluated.   Moreover, energy management strategies for pure electric vehicles and charging technologies are investigated.

Literature [25] is a study of braking strategy, which puts forward a braking strategy aimed at improving braking energy recovery efficiency and avoiding faults. The feasibility of the braking strategy is verified by co-simulation.

Literature [30] proposed a design method for driving the motor of the pure electric vehicle. Based on urban road conditions, the matching calculation method of driving motor parameters is studied. Simulation results show that the method is reasonable.

In this paper, we analyze the structural principle and dynamic characteristics of MEH-DCDS. The feasibility of MEH-DCDS was verified and installed on an electric vehicle to illustrate the feasibility of MEHPC-EV.

Point 5: References must be written according to the journal's policy.

Response 5: I used Zotero software to edit the format of the references.

Round 2

Reviewer 1 Report

The authors have addressed my comments.